# Activation Energy of SDS–Protein Complexes in Capillary Electrophoresis with Tetrahydroxyborate Cross-Linked Agarose Gels

**DOI:** 10.3390/gels10120805

**Published:** 2024-12-07

**Authors:** Dániel Sárközy, András Guttman

**Affiliations:** 1Horváth Csaba Memorial Laboratory of Bioseparation Sciences, Research Center for Molecular Medicine, Faculty of Medicine, Doctoral School of Medicine, University of Debrecen, 4032 Debrecen, Hungary; daniel.sarkozy@med.unideb.hu; 2Translational Glycomics Research Group, Research Institute of Biomolecular and Chemical Engineering, University of Pannonia, 8200 Veszprem, Hungary

**Keywords:** agarose gel, temperature, activation energy, SDS–protein complexes, capillary electrophoresis

## Abstract

Hydrogels like agarose have long been used as sieving media for the electrophoresis-based analysis of biopolymers. During gelation, the individual agarose strands tend to form hydrogen-bond mediated double-helical structures, allowing thermal reversibility and adjustable pore sizes for molecular sieving applications. The addition of tetrahydroxyborate to the agarose matrix results in transitional chemical cross-linking, offering an additional pore size adjusting option. Separation of SDS-proteins during gel electrophoresis is an activated process defined by the interplay between viscosity, gelation/cross-link formation/distortion, and sample conformation. In this paper, the subunits of a therapeutic monoclonal antibody were separated by capillary SDS agarose gel electrophoresis at different temperatures. The viscosity of the separation matrix was also measured at all temperatures. In both instances, Arrhenius plots were used to obtain the activation energy values. It was counterintuitively found that larger SDS–protein complexes required lower activation energies while their low-molecular-weight counterparts needed higher activation energy for their electromigration through the sieving matrix. As a first approximation, we considered this phenomenon the result of the electric force-driven distortion of the millisecond range lifetime reticulations by the larger and consequently more heavily charged electromigrating molecules. In the meantime, the sieving properties of the gel were still maintained, i.e., they allowed for the size-based separation of the sample components, proving the existence of the reticulations. Information about the activation energy sheds light on the possible deformation of the sieving matrix and the solute molecules. In addition, the activation energy requirement study helped in optimizing the separation temperature, e.g., with our sample mixture, the highest resolution was obtained for the high-molecular-weight fragments, i.e., between the non-glycosylated heavy chain and heavy-chain subunits at 25 °C (lower E_a_ requirement), while 55 °C was optimal for the lower-molecular-weight light chain and non-glycosylated heavy chain pair (lower E_a_ requirement). Future research directions and possible applications are also proposed.

## 1. Introduction

Agarose is known for its robust and rapid gelation via hydrogen bonds, resulting in firm and brittle gels when cooled under stable conditions [1]. Individual agarose strands are inclined to form double-helical structures during gel formation, imparting its characteristic thermal reversibility and adjustable pore sizes. The reversible gelation properties of agarose and the ability to form a sturdy sieving matrix at relatively low concentrations make it suitable for separating a broad range of biopolymers [2]. Therefore, it is widely used as an electrophoretic sieving medium, primarily in DNA and RNA but also in protein analysis [3,4]. At lower temperatures, agarose gels tend to become more rigid and less elastic as molecular mobility is restricted. In contrast, higher temperatures can increase network flexibility and weaken the physical cross-links, leading to a reduction in gel strength. Prolonged exposure to elevated temperatures may degrade agarose gels by breaking down the polymer chains, decreasing mechanical integrity, and causing irreversible structural changes [5]. These temperature effects can be unfavorable during slab gel electrophoresis due to uneven heating, which can lead to gel distortion, inconsistent pore sizes, and unpredictable shifts in analyte migration times. Using agarose gels in capillary electrophoresis, on the other hand, the precisely controlled separation temperatures can be beneficial by ensuring uniform heating along the capillary column, thus enhancing separation efficiency, and reproducibility. Consequently, capillary agarose gel electrophoresis (CAGE) further expands the utility of agarose-based separation matrices, utilizing closely controlled temperatures even at high separation voltages (up to 30 kV) and with minimized sample adsorption within narrow-bore capillaries (typically 50–100 μm i.d.).

As early as in the 1950s, the potential of agar-based gels was recognized for protein electrophoresis, yet the presence of agaropectin and other impurities initially led to sample adsorption and precipitation, which hindered separation power [6]. Later, it was demonstrated that purified agarose showed minimal adsorption and interaction with dyes, underscoring its applicability as an electrophoretic medium to achieve consistent and reproducible results for the size-based separation of polyionic biopolymers [7]. Understanding the relationship between the gel pore radius and agarose concentration allowed the creation of gel pore sizes in the low nanometer to micrometer scale, accommodating diverse electrophoretic applications and enabling the separation of analytes with a wide range of molecular sizes [8]. Ferguson plots were used to quantify the pore size variability across different agarose concentrations, providing a method to predict and customize gel properties for size-specific separations [9]. The robust gel structure of agarose provided a highly reproducible matrix for separating not only nucleic acids but SDS–protein complexes as well [10]. The addition of boric acid to agarose-based electrophoretic separation matrices facilitates specific pore size adjustments. The generation of tetrahydoxyborate ions in the aqueous medium allows the formation of reversible covalent bonds in the millisecond timescale with the vicinal hydroxyl groups of sugar-building block-based polymers [11] such as agarose. This cross-linking process enables the formation of a more structured matrix, with smaller pore sizes leading to the enhanced resolution of smaller molecules while introducing resistance to larger ones. Exploiting this option, the separation of immunoglobulin subunits with varying charge and size profiles was reportedly optimized using agarose–borate gels in SDS–capillary gel electrophoresis (SDS-CAGE) [12].

Activation energy is a critical factor in the context of temperature effects on hydrogels like agarose. It can be defined as the minimum amount of energy required for specific processes to occur, such as the gelation or the reversal of the gel structure. Performing capillary electrophoresis in tetrahydroxyborate cross-linked agarose gels, the activation energy has critical roles during gel formation, cross-linking, and the electromigration of the separating biopolymers. In other words, introducing tetrahydroxyborate ions into the agarose gel represents an additional complexity due to adduct formation, which is also responsible for the more structured gel matrix [13,14]. This influences the activation energy requirement both for viscous flow and matrix formation, as well as the electrophoretic movement and possible deformation of the sample molecules migrating through the reticulations, along with the efficiency and resolution of the separation.

In this paper, the activation energy requirement of different-size SDS–protein complexes is investigated by capillary electrophoresis using tetrahydroxyborate cross-linked agarose gels. The subunits of a therapeutic monoclonal antibody are separated at different temperatures, and Arrhenius plots are used to derive the activation energy values. The activation energy for the viscous flow is also determined to understand its contribution. The effects of the differential activation energy requirements on resolution are also discussed.

## 2. Results and Discussion

### 2.1. Theoretical Considerations

In aqueous solutions, tetrahydroxyborate is formed from boric acid, resulting in a positively charged hydroxonium ion, and, therefore, decreasing the pH.
B(OH)_3_ + 2H_2_O ⇌ B(OH)_4_^−^ + H_3_O^+^(1)

The tetrahydroxyborate anion can react with one or two diol groups of the agarose chains, forming 1:1 (Equation (2)) and 2:1 (Equation (3)) adducts.
B(OH)_4_^−^ + A ⇌ B(OH)_2_A^−^ + 2H_2_O(2)
B(OH)_2_A^−^ + A ⇌ BA_2_^−^ + 2H_2_O(3)

Cross-linking of the agarose chains is the result of the formation of 2:1 complexes, as shown in Figure 1. Please note that the formed adducts constantly disrupt and rebound in the millisecond time scale, resulting in a transient gel structure, due to their high-frequency lifetime [15].

Under the influence of an electric field, charged species migrate at different velocities due to their hydrodynamic volume-to-charge ratio, which is the basic principle of electrophoretic separation methods [16]. Applying a constant electric field (E), the electromigrating species with a Q net charge is exposed to an electric force (F_e_):(4)$$F_{e}=Q\cdot E$$

The solute molecules are also subject to a frictional force (F_f_) retarding their electromigration, expressed as the product of the translational friction coefficient (*f*) and the linear velocity:(5)$$F_{f}=f\cdot \left(\frac{dx}{dt}\right)$$
where *dx* represents the migration distance and *dt* is the migration time. If these two forces are in balance, the analyte molecules migrate at a constant velocity (*v*):(6)$$v=\frac{dx}{dt}=\frac{E\cdot \;Q}{f}$$

The term *f* is proportional to the viscosity of the separation medium (ⴄ) and also influenced by the molecular characteristics of the electromigrating species (Mw^k^), where Mw is the molecular weight and k~0.2 in SDS-CGE [3]; thus,
(7)$$v=E\;\frac{Q}{{Mw}^{k}\;\eta }$$

Considering the practically equal charge-to-mass ratio of the SDS–protein complexes, Equation (7) can be simplified by introducing Const_Q/Mw_, which represents the Q/Mw^k^ term as, this value would be the same for the proteins in the sample mixture, regardless of their size, due to the 1.4 g SDS/g protein complexation ratio.

The viscosity, on the other hand, can be defined by the modified Arrhenius equation [17,18] as follows:(8)$$ⴄ={Const}_{v}{\;\cdot e}^{E_{a}/RT}$$

Const_v_ represents a pre-exponential factor, E_a_ is the activation energy, R is the universal gas constant, and T is the absolute temperature. Combining Equations (7) and (8), and considering that the electrophoretic mobility (µ) is by definition the field strength normalized velocity, µ can be expressed as follows:(9)$$µ=\frac{v}{E}={Const}_{v}{\cdot Const}_{Q/Mw}\cdot e^{-E_{a}/RT}$$

Therefore, E_a_ can be readily calculated from the slopes of the logarithmic mobility vs. reciprocal absolute temperature plots (Arrhenius plots).

However, one should consider that the activation energy is the sum of the activation energy requirements [3] for the viscous flow (E_a_^v^), as well as the distortion of the dynamic pore structure of the gel (E_a_^g^) and the conformation changes of the electromigrating sample molecules (E_a_^s^). While E_a_^v^ can be readily measured (see Section 2.5 below), it is not possible to differentiate between the distortion effect on the sieving matrix and the deformation on the sample components, as was suggested earlier by Cottet and Gareil [19]. Therefore, we introduce E_a_^d^ (covering both matrix and sample) as the sum of E_a_^g^ and E_a_^s^, representing all the distortion/conformation changes.
E_a_ = E_a_^v^ + E_a_^g^ + E_a_^s^ = E_a_^v^ + E_a_^d^
(10)

### 2.2. Reproducibility Study

Tetrahydroxyborate cross-linked agarose gels are special hydrogels combining the physical gel characteristics of agarose and the chemical gel attributes of the transient cross-linking between agarose polymer chains. In other words, tetrahydroxyborate cross-linking results in reversible covalent bonds with the hydroxyl groups on the agarose chains, creating a highly structured sieving medium. However, considering the possible effects of the millisecond-range disruption and rebounding of the sieving matrix, prior to the study on the activation energy requirement, it was important to understand the reproducibility of the capillary SDS agarose–borate gel-based separation of the sample mixture components containing the 10 kDa internal protein standard, as well as the light chain (LC, Mw 23.89 kDa), non-glycosylated heavy chain (ngHC, Mw 47.87 kDa) and heavy chain (HC, Mw 49.37 kDa) subunits of the therapeutic monoclonal antibody of omalizumab. The overlay of ten consecutive separations of the LC, ngHC, and HC subunits of the monoclonal antibody sample is shown in Figure 2, along with the 10 kDa internal protein standard. As one can observe, excellent overlap was obtained among the traces, suggesting the very predictive forming and re-forming of reticulation structures despite the transient characteristics of the matrix. Table 1 presents a statistical analysis of the migration time and peak area percentage repeatability, confirming consistent results across the ten parallel runs displayed in Figure 2. Peak alignment of the ten consecutive runs showed migration time and peak area percentage RSD values below 0.6% and 6%, respectively.

### 2.3. Temperature Dependence of the Separation in Borate Cross-Linked Agarose Gels

Based on the encouraging, highly reproducible results shown above, we confidently started the activation energy study on the separation of SDS–protein complexes in capillary electrophoresis using tetrahydroxyborate cross-linked agarose gels. This investigation was based on the evaluation of the effect of temperature, because of its significant role in the evolution of the sieving properties of tetrahydroxyborate-agarose gels. While agarose gelation requires lower temperatures, the simultaneous high-frequency formation of the 2:1 tetrahydroxyborate–agarose adducts (Figure 1) facilitates the transitional cross-linking of the resulting matrix. Therefore, the activation energy requirement of such gels is supposedly greater than that for agarose only, due to the additional thermal energy needed for cross-link formation (E_a_^g^ in Equation (10)). This transient cross-linking results in a slightly higher gelation temperature but better thermal stability due to the more oriented arrangements. Figure 3 compares the separations of the SDS–protein test mixture components at different capillary temperatures between 25 °C and 55 °C in 10 °C intervals. An increase in temperature generally decreases the viscosity of gels as thermal energy reduces the intermolecular forces holding the gel structure together, i.e., changing the activation energy requirement of the viscous fluid (E_a_^v^; see Section 2.5 below). Thus, as expected, the migration times of all sample components decreased with increasing capillary temperature.

To shed light on the activation energy requirement for the electromigration of the different-molecular-weight proteins in the sample mixture, Arrhenius plots were constructed using the natural logarithms of electrophoretic mobilities, derived from the migration times of the SDS–protein complexes in Figure 3. Please note that the electrophoretic mobility values of the sample components were adjusted by considering the temperature-dependent viscosity alteration of the sieving matrix (4% for every 5 °C increase) [20]. Figure 4 shows the resulting Arrhenius plots of logarithmic mobility vs. reciprocal absolute temperature.

### 2.4. The Activation Energy Plot

It can be seen from the slopes in Figure 4 that the plots are non-parallel, suggesting that the activation energy for the electromigration of the sample components in tetrahydroxyborate cross-linked agarose gels was molecular weight-dependent. According to Equation (9), multiplying the slopes of the log µ vs. 1/T plots with the universal gas constant, one can obtain the activation energy values required for the sample components to pass through the reticulations of the tetrahydroxyborate–agarose gel matrix under the influence of the applied electric field. The resulting activation energy vs. molecular weight plots of the separated SDS–protein complexes are shown in Figure 5.

As one can observe in Figure 5, the activation energy is a logarithmically decreasing function of the molecular weight (r^2^ = 0.994), contradicting the general assumption of a greater E_a_ requirement for the electromigration of larger molecules in chemically cross-linked gels [21]. Considering the fact that the tetrahydroxyborate anions only transiently cross-link the agarose chains in a millisecond timescale, this feature can somewhat ease the passage for the larger proteins via distortion during the non-chemically cross-linked stage through the still existing but relaxed reticulations, i.e., when the agarose strands are bound only with physical interactions, similar to what has been observed earlier with DNA sequencing fragment separation in non-cross-linked polyacrylamide gels [22]. This effect is probably mediated by the electrostatic force (Equation (4)); in other words, larger but otherwise similar charge per mass ratio proteins are more energetic for distorting the transitional gel structure, requiring a lower activation energy (E_a_^g^) to pass through the reticulations, resulting in the observed results. Again, it is important to emphasize that this phenomenon did not affect the general sieving effect of the agarose gel, as evidenced by Figure 2 and Figure 3. In other words, distortion of the matrix and deformation of the sample components only decreased the activation energy requirement, but did not affect the molecular-sieving-based size separation. However, the resolution between the sample components changed with the separation temperature. The highest resolution between the lower-molecular-weight LC and ngHC subunits was obtained at 55 °C (Rs = 3.58 at 55 °C vs. Rs = 3.46 at 25 °C), probably due to the larger activation energy drop of 1.03 kJ/mol (Figure 5, LC vs. ngHC). In contrast, the non-glycosylated heavy-chain and higher-molecular-weight heavy-chain pair featured the best separation at 25 °C (Rs = 1.56 at 25 °C vs. Rs = 1.20 at 55 °C), in which case the activation energy difference was only 0.24 kJ/mol (Figure 5, ngHC vs. HC).

### 2.5. Temperature-Dependent Viscosity of the Borate Cross-Linked Agarose Gels

Considering Equation (10), it was important to investigate the viscosity changes of the sieving medium as a function of the temperature to obtain the corresponding E_a_^v^ value, and to calculate the activation energy requirement for the distortion/conformation changes (E_a_^d^). Based on Equation (8), the logarithmic viscosity vs. reciprocal absolute temperature relationship was plotted, as shown in Figure 6. The activation energy for the viscous flow was then calculated from the slope by multiplying it with the universal gas constant (R), resulting in E_a_^v^ = 14.59 kJ/mol.

Taking into consideration the above-derived E_a_^v^ value, we calculated the distortion/conformation-change-related activation energy requirements (E_a_^d^ = E_a_ − E_a_^v^, Table 2) for the sample components based on the data in Figure 5 and obtained 25.17, 23.92, 22.89 and 22.66 kJ/mol for the 10 kDa internal standard, as well as the LC, ngHC and HC subunits of the monoclonal antibody sample, respectively. These results suggested that approximately 60% of the total activation energy requirement for the electromigration of the analyte components in capillary electrophoresis using borate cross-linked agarose gels was conformation-change related, i.e., a result of the combined effects of the sieving matrix distortion and the molecular shape deformation of the sample, as shown in Table 2. The remaining ~40% of the activation energy was required to prevail the energy barrier that was represented by the viscosity of the sieving matrix.

## 3. Conclusions

The activation energy concept in capillary gel electrophoresis refers to the energy required to overcome the frictional resistance of the sieving matrix, which correlates with the hydrodynamic volume of the electrophoretically migrating but similar mass-to-charge ratio molecular species. In general, in permanently cross-linked matrices, such as the broadly used polyacrylamide/bis-acrylamide gels, the larger molecules face more resistance during their electromigration via the reticulations due to the greater and not deformable physical obstruction compared to smaller molecules, which on the other hand can cross through the interconnected channels of the matrix more easily because of the lower physical resistance, consequently needing lower activation energy. In our study, we investigated the activation energy requirement for the electromigration of SDS-bound monoclonal antibody fragments in tetrahydroxyborate cross-linked agarose gels. Contrary to the general assumption, we observed that larger SDS–protein complexes required lower activation energies for the electrophoresis process than that required for lower-molecular-weight ones, i.e., E_a_: 10 kDa > LC (23.89 kDa) > ngHC (47.87 kDa) > HC (49.37 kDa) (Table 2). This phenomenon was considered the result of the electric force-driven distortion of the millisecond-range lifetime sieving network during the non-cross-linked stage of the matrix. It is noteworthy that the molecular sieving properties of the gel were still maintained, i.e., they allowed for adequate size-based separation of all the SDS–protein complexes in the sample mixture. More importantly, information about the activation energy requirement for the sample components to pass through the gel matrix helped to optimize the temperature of the separation to attain the required resolution between the concerned sample components. It was especially apparent in the resolution between the lower Mw LC and ngHC chain subunits, in which case the highest resolving power was obtained at 55 °C with a 1.03 kJ/mol activation energy drop. This contrasted the non-glycosylated heavy-chain and the higher-Mw heavy-chain pair where a 25 °C separation temperature was optimal for their highest-resolution separation with only a 0.24 kJ/mol differential activation energy requirement. Consequently, a greater activation energy difference between sample components required a higher separation temperature for better resolution. In addition, the activation energy requirement for the viscous flow (E_a_^v^) was also measured and found to be ~40% of the total E_a_ needed for the electromigration of the sample components. Information about activation energy requirements sheds light on the possible distortion of the sieving matrix and the deformation of the solute molecules. Nevertheless, it was reasonable to consider the opposite, i.e., changes in the activation energy may reflect deformation in the gel structure or analyte shape, especially considering the >60% share of E_a_^d^ in the total E_a_ requirement (Equation (10) and Table 2).

Future research directions with tetrahydroxyborate cross-linked agarose gels should be based on the utilization of reversible cross-linking to fine-tune the mechanical properties of the resulting gel matrices by adjusting the agarose concentration and chain length, and the boric acid concentration, as well as the pH and the ionic strength of the background electrolyte. In addition to separation optimization, the information obtained could be useful for controlled release applications or the recovery of substances trapped in a gel matrix. For example, it would be of high importance in immunoaffinity-based analyses when specific antibodies entrapped in a gel matrix are utilized to capture their antigens and released when all other sample components have migrated out of the separation platform, not interfering with the detection. The same principle can be readily applied in special bioengineering and drug delivery applications, in which instance, controlled release can be beneficial. Also, the fine-tuned mechanical properties of tetrahydroxyborate cross-linked agarose matrices can be used as tissue engineering scaffolds. The development of special highly cross-linked gel formulations would enable the high-resolution analysis of small nucleic acids such as micro RNS molecules, whose high importance was recently discovered [23]. On the other hand, gel–buffer formulations can also be optimized for the separation of large protein complexes, considering the fact that the activation energy requirement for the electromigration of those species is smaller.

## 4. Materials and Methods

### 4.1. Chemicals and Reagents

Agarose (ultra-low gelation temperature), HPLC grade water, boric acid, sodium lauryl sulfate (SDS), EDTA·Na_2_, Tris, NaOH, HCl, glycerol, and 2-mercaptoethanol were from Sigma Aldrich (St. Louis, MO, USA) were used. The 10 kDa internal standard protein and the sample buffer (SDS-MW) were from Beckman Coulter (Brea, CA, USA). The endoglycosidase (PNGase F) was kindly provided by the Bio-Nanosystems Laboratory of Pannon University (Veszprem, Hungary). The monoclonal antibody therapeutic (Xolair) omalizumab was donated by the Semmelweis Hospital (Miskolc, Hungary).

### 4.2. Agarose Gel and Sample Preparation

The Tris-borate-EDTA-glycerol (TBEG) background electrolyte for the agarose gel contained 4% (*w*/*v*) boric acid, and was set to pH 8.0 with Tris-base, before adding EDTA·Na_2_ and glycerol in a 2 mM and 10% (*v*/*v*) final concentration, respectively. Please note that the addition of glycerol shifted the pH to 7.0. The agarose was then added in a 0.8% (*w*/*v*) final concentration and stirred overnight at 75 °C on a magnetic hotplate (250 RPM), followed by the addition of 0.2% (*w*/*v*) SDS and slow mixing at room temperature (100 RPM) for an hour to avoid foaming.

The sample was prepared by adding 10 μL of 10 mg/mL anti-asthma therapeutic monoclonal antibody (omalizumab, also known as Xolair) to 80 μL of 100 mM Tris-HCl, with 1% SDS (pH 9.0, referred to as sample buffer). Then, 2 μL of the 10 kDa internal standard solution and 5 μL of 2-mercaptoethanol (2-ME) were added into the PCR tube, making up a 1 mg/mL sample stock solution. The use of 2-ME was essential as it reduced the disulfide bonds in the antibody, promoting denaturation and ensuring the unfolding of the protein structure, which was also necessary to facilitate the subsequent enzymatic deglycosylation reaction. In other words, breaking the disulfide bonds enhanced the accessibility of the glycosylation sites of the antibody, thus enabling more efficient enzymatic removal of the asparagine-linked glycans. For the removal of N-glycans from the heavy-chain subunit of the therapeutic monoclonal antibody, the sample was denatured by applying a temperature gradient protocol to avoid artifacts due to possible fragmentation. Utilizing the 2-ME, Tris-HCl and SDS-containing sample buffer for denaturation, the temperature was raised at a rate of five degrees centigrade per minute from 30 °C to 80 °C, followed by five-minute isothermal incubation at 80 °C. After the denaturation step, endoglycosidase digestion was initiated by adding 2.0 µL of the PNGase F (200 mU) to the reaction mixture, followed by incubation at 50 °C for an additional hour to ensure full removal of N-linked glycan structures. The enzymatic reaction of PNGase F specifically cleaved the link between the asparagine residues of the polypeptide backbone and the attached N-glycans, thereby releasing them from the HC fragment of the mAb [24].

### 4.3. SDS–Capillary Agarose Gel Electrophoresis (SDS-CAGE)

In all SDS-CAGE analyses, a P/ACE MDQ Capillary Electrophoresis System (Beckman Coulter, Brea, CA, USA) was used with ultraviolet absorbance detection (214 nm), employing a 20 cm effective length (50 μm i.d.) bare fused silica capillary (total length 30 cm). Capillary conditioning included rinsing with 0.1 M sodium hydroxide for three minutes, 0.1 M hydrochloric acid for one minute, and HPLC-grade water for four minutes before their first use. Prior to each run, the capillary was filled with the agarose gel–buffer system by applying a 80 psi rinse pressure for 4 min. The capillary conditioning steps were repeated after every three runs. Samples were injected electrokinetically for 10 s at −5 kV. The separations were performed with the cathode at the injection end (reversed-polarity mode) with a −15 kV electric potential at 25, 35, 45, and 55 °C. The separation temperatures were precisely controlled by the liquid-cooling system of the capillary electrophoresis device, with ±0.1 °C accuracy. Data processing and analysis were performed using the 32Karat software (version 10.1, Sciex). For manual corrections and peak area determination (i.e., integration), the Log4 Normal distribution model was utilized with r^2^ ≥ 0.99.

## Figures and Tables

**Figure 1 gels-10-00805-f001:**
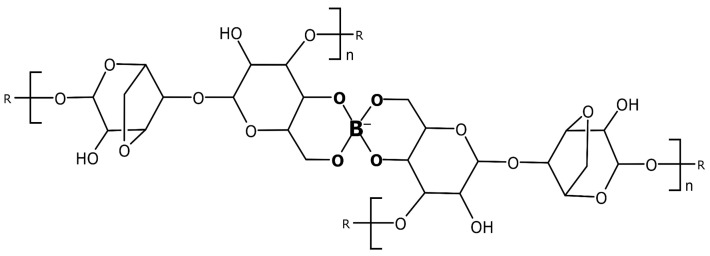
Schematic representation of the di–diol (2:1) linkage formation during agarose gel cross-linking with tetrahydroxyborate.

**Figure 2 gels-10-00805-f002:**
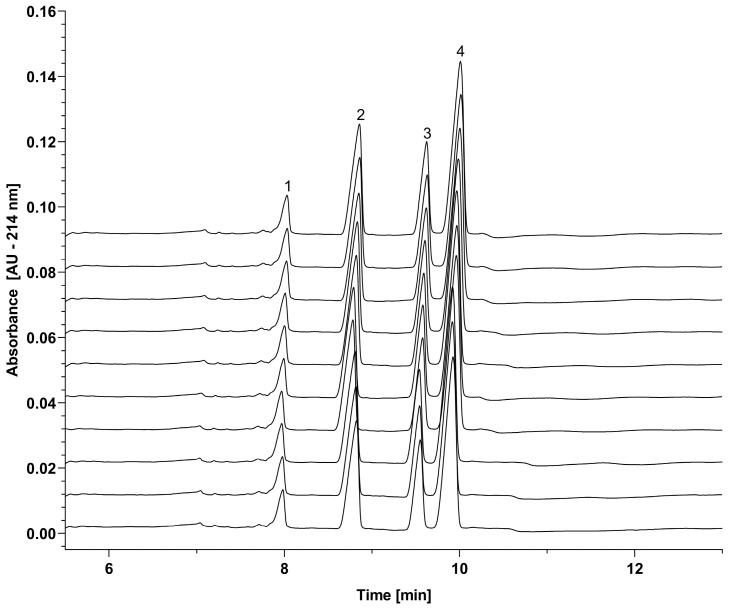
Reproducibility of SDS capillary agarose gel electrophoresis separation of the subunits of omalizumab using tetrahydroxyborate cross-linker. The overlay of ten consecutive runs demonstrates reproducible peak alignment, with all peaks showing consistent migration times and peak shapes. Peaks: (1) 10 kDa internal standard, (2) light-chain subunit, (3) non-glycosylated heavy-chain subunit, and (4) heavy-chain subunit of omalizumab. Conditions: sieving matrix, 0.8% (*w*/*v*) agarose in TBEG background electrolyte (pH 7.0); capillary, 20 cm effective length 50 µm i.d. bare fused silica (total length: 30 cm); detection, 214 nm ultraviolet light absorption; applied voltage, −15 kV (anode at the detection side); separation temperature, 25 °C ± 0.1 °C; electrokinetic injection, −5 kV for 10 s.

**Figure 3 gels-10-00805-f003:**
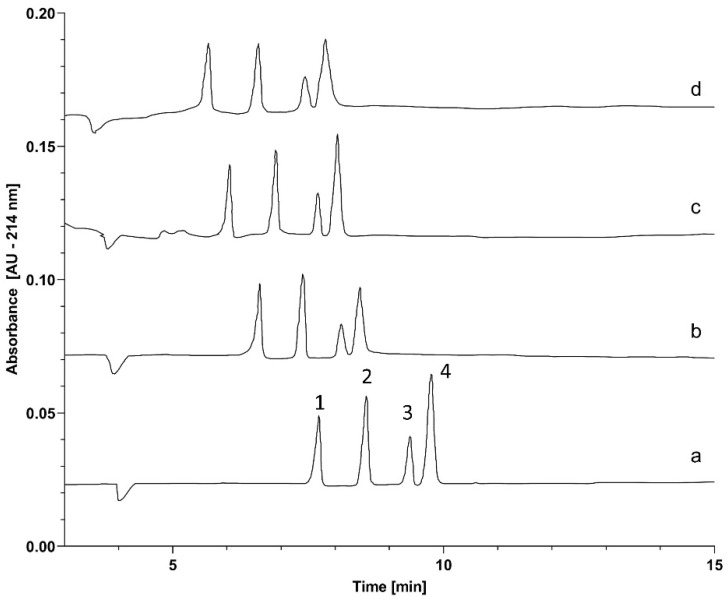
The effect of temperature on the SDS capillary agarose gel electrophoresis separation of the subunits of the therapeutic monoclonal antibody omalizumab, using tetrahydroxyborate cross-linker. Traces: (**a**) 25 °C, (**b**) 35 °C, (**c**) 45 °C and (**d**) 55 °C. Peaks: (1) 10 kDa internal standard, (2) LC, (3) ngHC, and (4) HC fragments. Separation conditions are the same as in Figure 2, except for the temperatures of the analyses.

**Figure 4 gels-10-00805-f004:**
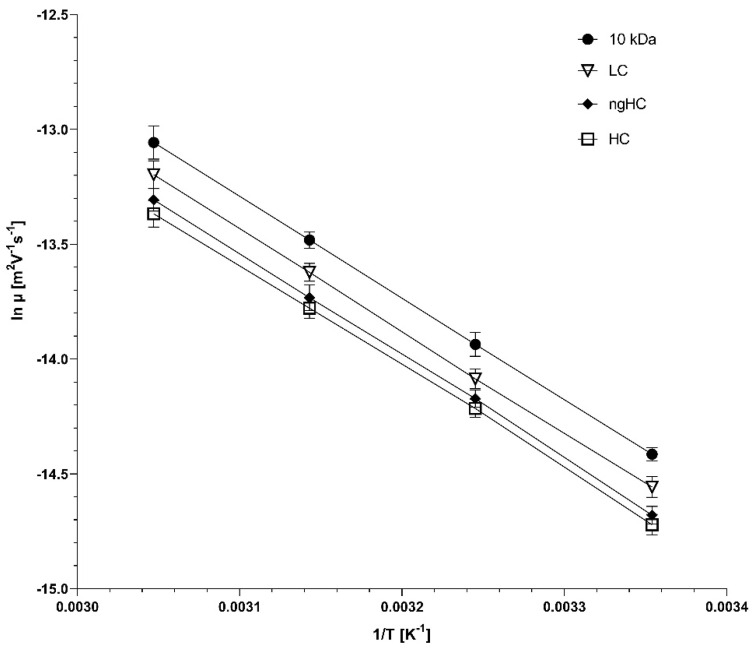
Arrhenius plots of logarithmic mobility vs. reciprocal absolute temperature of the SDS–protein complexes in the sample mixture. The symbols representing the sample molecules are defined in the inset. The error bars show the standard deviation of 3 parallel experiments.

**Figure 5 gels-10-00805-f005:**
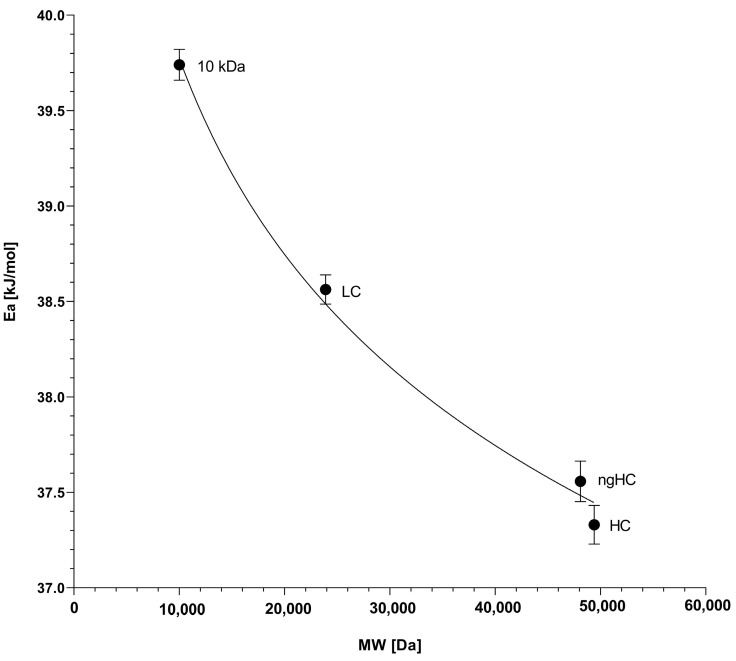
Relationship between the activation energies (derived from the slopes of Figure 4) and the molecular weights of the separated SDS–protein complexes. The error bars represent the standard deviation of 3 parallel experiments.

**Figure 6 gels-10-00805-f006:**
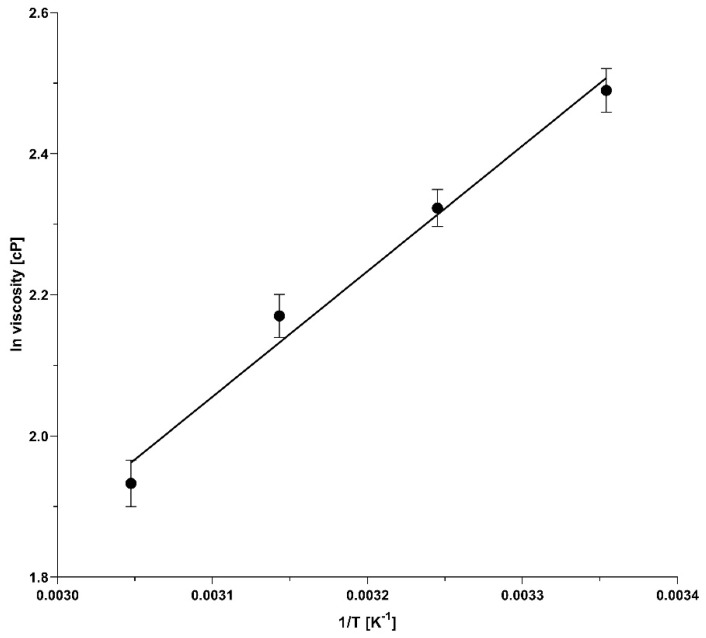
Logarithmic viscosity vs. reciprocal absolute temperature (Arrhenius) plot representing the temperature-dependent viscosity changes of the borate cross-linked agarose gels (r^2^ = 0.984).

**Table 1 gels-10-00805-t001:** Statistical evaluation of the migration time and peak area percentage repeatability of ten consecutive runs using 0.8% (*w*/*v*) agarose–TBEG gel buffer system.

Run	LC		ngHC		HC	
	Migration Time [min]	Peak Area %	Migration Time [min]	Peak Area %	Migration Time [min]	Peak Area %
**1**	8.832	27.689	9.564	17.495	9.936	47.537
**2**	8.814	28.973	9.541	16.826	9.921	47.313
**3**	8.765	28.047	9.503	16.422	9.877	48.232
**4**	8.878	28.425	9.679	19.094	10.054	45.595
**5**	8.893	27.442	9.687	19.341	10.071	46.397
**6**	8.804	27.069	9.578	18.290	9.954	48.052
**7**	8.827	28.786	9.601	18.535	9.977	45.762
**8**	8.827	27.521	9.605	19.362	9.991	45.681
**9**	8.842	27.763	9.614	17.931	9.996	47.377
**10**	8.841	27.943	9.611	19.294	9.992	45.649
**Average**	8.832	27.966	9.598	18.259	9.977	46.760
**%RSD**	**0.386**	**2.054**	**0.560**	**5.572**	**0.558**	**2.139**

**Table 2 gels-10-00805-t002:** Activation energy values for the electromigration of the sample components considering the effects of the viscous flow and the matrix/solute distortion/conformation changes.

Activation Energy (kJ/mol)	10 kDa	LC (23.89 kDa)	ngHC (47.87 kDa)	HC (49.37 kDa)
E_a_ total activation energy	39.75	38.50	37.48	37.24
E_a_^v^ (viscous flow)	14.59	14.59	14.59	14.59
E_a_^d^ (distortion /conformation)	25.17	23.92	22.89	22.66

## Data Availability

The data presented in this study are available on request from the corresponding author.

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
