# Peer review of "Activation Energy of SDS–Protein Complexes in Capillary Electrophoresis with Tetrahydroxyborate Cross-Linked Agarose Gels"

_gels, 2024, doi:10.3390/gels10120805_

Round 1

Reviewer 1 Report

Comments and Suggestions for Authors

This is a well organized and referenced manuscript reporting and discussing the separation of the subunits of a therapeutic monoclonal antibody by capillary sodium dodecyl sulfate gel electrophoresis, using tetrahydroxyborate cross-linked agarose gels at different temperatures.  The experimental session is logically designed and clearly described and the discussion comprehensibly explains the meaning of the results. These include the calculation of the activation energy associated to the electromigration process of the SDS-protein complexes in tetrahydroxyborate cross-linked agarose gels that the manuscript demonstrates to be lower for larger SDS-protein complexes than that of the lower molecular mass protein subunits.  In conclusion, the submitted manuscript is within the scope of “Gels”, reports valuable results of an interesting study and should be considered for publication as it is.  

Author Response

REVIEWER 1

This is a well organized and referenced manuscript reporting and discussing the separation of the subunits of a therapeutic monoclonal antibody by capillary sodium dodecyl sulfate gel electrophoresis, using tetrahydroxyborate cross-linked agarose gels at different temperatures.  The experimental session is logically designed and clearly described and the discussion comprehensibly explains the meaning of the results. These include the calculation of the activation energy associated to the electromigration process of the SDS-protein complexes in tetrahydroxyborate cross-linked agarose gels that the manuscript demonstrates to be lower for larger SDS-protein complexes than that of the lower molecular mass protein subunits.  In conclusion, the submitted manuscript is within the scope of “Gels”, reports valuable results of an interesting study and should be considered for publication as it is.  

REPLY: We greatly appreciate the reviewer’s suggestion to accept our paper as is.

Reviewer 2 Report

Comments and Suggestions for Authors

Summary

Sárközy et al explored the activation energy required for the migration of SDS-protein complexes in capillary electrophoresis (CE) using tetrahydroxyborate cross-linked agarose gels. Using the subunits of omalizumab, a monoclonal antibody, the authors determined the activation energies of the subunits, and they found that larger SDS-protein complexes exhibit lower activation energy than smaller ones. The authors attributed this to distortions in the gel's transient cross-linked structure under an applied electric field, favoring the migration of larger complexes.

The authors present their findings clearly and provide valuable insights into the interplay between gel structure, electrophoretic conditions, and activation energy in the context of tetrahydroxyborate cross-linked agarose gels. This work is highly relevant to the readership of Gels. I recommend the manuscript for publication, either as is or with minor revisions.

Major concerns:

1.    Although the authors already mentioned adjusting boric acid concentration as one of their future research directions, I was wondering if the authors could elaborate more on how this would impact the relationship between activation energy and molecular weight of the analytes. Specifically, given that the study demonstrates activation energy as a logarithmically decreasing function of molecular weight, could higher borate concentrations reduce the occurrence of non-cross-linked stages in the gel matrix? If so, might this stabilization reverse the observed trend, resulting in a different activation energy versus molecular weight relationship? Additional insights into this mechanism would enhance the understanding of borate’s role in shaping electrophoretic behavior.

2.    In the Theory section, the authors present the equation 𝑣=𝐸𝑄/𝑀𝑤𝑘𝜂, where k ~ -0.2. This implies that a higher molecular weight 𝑀𝑤 results in a lower 𝑀𝑤𝑘 and concurrently, the manuscript indicates that higher 𝑀𝑤 corresponds to lower activation energy (Ea), which leads to a lower viscosity (𝜂) as =𝑐𝑜𝑛𝑠𝑡 𝑒𝐸𝑎/𝑅𝑇. Additionally, high molecular weight species are likely to carry more charge (Q), which suggests that their electrophoretic velocity should be higher. However, despite this, these species with high 𝑀𝑤 are observed to elute slower. Could the authors provide further insight into the interplay between molecular weight, charge, viscosity, and migration behavior in this system? The authors also mentioned the activation energy phenomenon did not affect the general sieving properties of the gel, can the author discuss more on how activation energy and sieving properties of the gel influence the separation of their analytes.

Minor comments:

1.    Although the figures in the manuscript file contain error bars, they are missing in the original images.

Author Response

Please see the details in the attachment.

Reviewer 3 Report

Comments and Suggestions for Authors

• What is the main question addressed by the research?

The main question is evaluation of activation energy of protein separation in dynamically cross-linked agarose in capillary electrophoresis.

• Do you consider the topic original or relevant to the field? Does it address a specific gap in the field? Please also explain why this is/ is not the case.

Topic is relevant since the authors studied the fundamental aspects of capillary electrophoresis, for first time focusing on the use of dynamically cross-linked agarose as the gel phase. This is indeed a significant fundamental finding for understanding the mechanisms of capillary electrophoresis

• What does it add to the subject area compared with other published material?

For first time, SDS-protein separation energy activation in dynamically cross-linked agarose was evaluated.

• What specific improvements should the authors consider regarding the methodology? What further controls should be considered?

The stated observation (in the case of agarose dynamically cross-linked with boric acid, larger SDS-proteins require lower activation energies for the electrophoresis process compared to those with lower molecular weights) is insufficient to support a full paper in a Q1-ranked journal. I would recommend that the authors either modify the manuscript into a short communication or provide additional experimental data, such as thermodynamic parameters (changes in enthalpy, entropy, and Gibbs free energy for separation process) or influence of other dynamic cross-linker on activation energy.

• Are the conclusions consistent with the evidence and arguments presented and do they address the main question posed? Please also explain why this is/is not the case.
Yes

• Are the references appropriate?

Yes
• Any additional comments on the tables and figures.

No

Author Response

  • What is the main question addressed by the research?

The main question is evaluation of activation energy of protein separation in dynamically cross-linked agarose in capillary electrophoresis.

REPLY: We thank the reviewer for this excellent summary of the main question of our research, indeed this is the first systematic study on the electromigration of SDS-proteins in agarose gel-filled capillary columns under high electric field strength as a function of temperature.

  • Do you consider the topic original or relevant to the field? Does it address a specific gap in the field? Please also explain why this is/ is not the case.

Topic is relevant since the authors studied the fundamental aspects of capillary electrophoresis, for first time focusing on the use of dynamically cross-linked agarose as the gel phase. This is indeed a significant fundamental finding for understanding the mechanisms of capillary electrophoresis

REPLY: We greatly appreciate this comment of the reviewer.

  • What does it add to the subject area compared with other published material?

For first time, SDS-protein separation energy activation in dynamically cross-linked agarose was evaluated.

REPLY: This recognition from the reviewer, i.e., being the first on this part of the field is greatly appreciated.

  • What specific improvements should the authors consider regarding the methodology? What further controls should be considered?

The stated observation (in the case of agarose dynamically cross-linked with boric acid, larger SDS-proteins require lower activation energies for the electrophoresis process compared to those with lower molecular weights) is insufficient to support a full paper in a Q1-ranked journal. I would recommend that the authors either modify the manuscript into a short communication or provide additional experimental data, such as thermodynamic parameters (changes in enthalpy, entropy, and Gibbs free energy for separation process) or influence of other dynamic cross-linker on activation energy.

REPLY: In this paper, as the reviewer has recognized, we wanted to publish our initial findings on the topic that is highly relevant since we studied the fundamental aspects of capillary electrophoresis, for the first time, focusing on the use of transitionally cross-linked agarose as the gel phase, a significant fundamental finding for understanding the mechanisms of capillary agarose gel electrophoresis of SDS-proteins. To comply with the reviewer’s suggestion to explain the phenomenon behind our observation, we did additional experiments to determine the temperature dependence of the viscosity and the associated activation energy requirement to be able to calculate the matrix deformation/sample conformation change based components of Ea at the various Mw levels (Table 2). Accordingly, we extended the Results and Discussion as well as the Conclusions sections by providing additional information about the observed phenomenon including information about activation energy distribution between viscosity and matrix distortion/sample conformational changes to shed light on the level of distortion of the sieving matrix and deformation the solute molecules during the electromigration process. Moreover, considering the excellent recommendation of the reviewer we plan accordingly for our next paper to investigate the actual thermodynamic parameters including changes in enthalpy, entropy, and Gibbs free energy for the separation process as well as the influence of other dynamic cross-linkers on activation energy.

  • Are the conclusions consistent with the evidence and arguments presented and do they address the main question posed? Please also explain why this is/is not the case.
    Yes
  • Are the references appropriate?

Yes

  • Any additional comments on the tables and figures.

No

With all of the changes and required additions we hope that our manuscript is acceptable for publication in GELS.

Your kind attention to this matter is greatly appreciated,

Andras Guttman

Round 2

Reviewer 2 Report

Comments and Suggestions for Authors

The authors addressed my comments. I recommend publishing this manuscript.

Reviewer 3 Report

Comments and Suggestions for Authors

The authors enhanced a number of studies in this work, so manuscript can be accepted